# Targetable Molecular Alterations in the Treatment of Biliary Tract Cancers: An Overview of the Available Treatments

**DOI:** 10.3390/cancers15184446

**Published:** 2023-09-06

**Authors:** Marine Valery, Damien Vasseur, Francesco Fachinetti, Alice Boilève, Cristina Smolenschi, Anthony Tarabay, Leony Antoun, Audrey Perret, Alina Fuerea, Thomas Pudlarz, Valérie Boige, Antoine Hollebecque, Michel Ducreux

**Affiliations:** 1Medical Oncology Department, Gustave Roussy, F-94805 Villejuif, France; alice.boileve@gustaveroussy.fr (A.B.); cristina.smolenschi@gustaveroussy.fr (C.S.); anthony.tarabay@gustaveroussy.fr (A.T.); leony.antoun@gustaveroussy.fr (L.A.); audrey.perret@gustaveroussy.fr (A.P.); alinacristina.fuerea@gustaveroussy.fr (A.F.); thomas.pudlarz@gustaveroussy.fr (T.P.); valerie.boige@gustaveroussy.fr (V.B.); antoine.hollebecque@gustaveroussy.fr (A.H.); michel.ducreux@gustaveroussy.fr (M.D.); 2Medical Biology and Pathology Department, Gustave Roussy, F-94805 Villejuif, France; damien.vasseur@gustaveroussy.fr; 3Dana-Farber Institute, Lowe Center for Thoracic Oncology, Boston, MA 02215, USA; francesco.fachinetti@gustaveroussy.fr; 4Université Paris-Saclay, Gustave Roussy, Inserm Unité Dynamique des Cellules Tumorales, F-94805 Villejuif, France; 5Département d’Innovation Thérapeutique, Gustave Roussy, F-94805 Villejuif, France

**Keywords:** biliary tract cancers, molecular targetable alterations, targeted therapies, *FGFR2* fusion, *IDH1* mutation

## Abstract

**Simple Summary:**

Biliary tract cancers (BTCs) are rare tumours associated with poor prognosis. In advanced-stage disease, two treatment lines are approved: gemcitabine-cisplatin in combination with checkpoint inhibitors, and 5-FU-oxaliplatin, with a median overall survival of approximately one year. About 40% of BTC bear a targetable molecular alteration, the most frequent being FGFR2 fusions and IDH1 mutations in about 15% of intrahepatic cholangiocarcinomas. In this review, we will describe the different molecular targetable alterations in BTC (including *FGFR2* and *NTRK* fusions, *IDH1* mutation, *BRAF*V600E mutation, *HER2* amplification, *BRCA1/2* mutation and *KRAS*G12C mutation), the targeted therapies that these patients can receive and the expected benefit.

**Abstract:**

Biliary tract cancers (BTCs) are rare tumours, most often diagnosed at an unresectable stage, associated with poor prognosis, with a 5-year survival rate not exceeding 10%. Only first- and second-line treatments are well codified with the combination of cisplatin-gemcitabine chemotherapy and immunotherapy followed by 5-FU and oxaliplatin chemotherapy, respectively. Many studies have shown that BTC, and more particularly intrahepatic cholangiocarcinoma (iCCA), have a high rate of targetable somatic alteration. To date, the FDA has approved several drugs. Ivosidenib targeting IDH1 mutations, as well as futibatinib and pemigatinib targeting *FGFR2* fusions, are approved for pre-treated advanced CCA. The combination of dabrafenib and trametinib are approved for BRAFV600E mutated advanced tumours, NTRK inhibitors entrectinib and larotrectinib for tumours bearing NTRK fusion and prembrolizumab for MSI-H advanced tumours, involving a small percentage of BTC in these three settings. Several other potentially targetable alterations are found in BTC, such as *HER2* mutations or amplifications or *KRAS*G12C mutations and mutations in genes involved in DNA repair mechanisms. This review aims to clarify the specific diagnostic modalities for gene alterations and to summarize the results of the main trials and developments underway for the management of advanced BTC with targetable alterations.

## 1. Introduction 

Biliary tract cancers (BTCs) are rare malignant tumours with poor prognosis, diagnosed at advanced and unresectable stages in 70% of cases. BTCs are subdivided into intrahepatic cholangiocarcinoma (iCCA) (20%), upstream of the second division of the bile ducts, extrahepatic cholangiocarcinoma (eCCA)—peri-hilar (60%) or distal (20%)—and gallbladder cancers. The incidence of cholangiocarcinoma (CCA) is approximately 0.35–2 cases per 100,000 people in Western countries. In Asia, the incidence is much higher due to the high incidence of viral (for iCCA) and chronic parasitic (for eCCA) hepatitis. The incidence of iCCA is increasing in Western countries, probably due to lifestyle changes and the increasing incidence of cirrhosis due to metabolic syndrome [1,2,3].

The treatment of advanced or metastatic BTC relies on two standards, cisplatin-gemcitabine-durvalumab [4] and 5-fluorouracil-oxaliplatin [5], as first- and second-line treatments, respectively. Another option has emerged with the very recent KEYNOTE-966 study, the combination of cisplatin-gemcitabine and pembrolizumab [6]. Nevertheless, the median overall survival of patients with unresectable cholangiocarcinoma is around 1 year, and the 5-year survival does not exceed 10% [7]. The development of high-throughput sequencing techniques and the widespread use of these techniques in routine care showed the high frequency of targetable molecular alterations in BTC [8,9]. Overall, 80% of BTC bears molecular alterations, with *TP53* and *KRAS* mutations being the most common [8,10,11,12]. In iCCA, 50% of the molecular abnormalities make patients eligible for specific treatment, including *FGFR2* fusions and *IDH1* mutations [13,14]. Therefore, the European Society for Medical Oncology (ESMO) recommends multi-gene sequencing (NGS) to detect ESCAT (ESMO Scale for Clinical Actionability of molecular Targets) level I genomic alterations in advanced cholangiocarcinoma as early as the first line of therapy [15,16] (including *IDH1* mutations, *FGFR2* fusions and *MSI-H* status). 

Several nucleic-acid-based assays are available for detecting molecular alterations in BTC. DNA-based NGS assays offer the advantage of high sensitivity for detecting single-nucleotide variants (SNVs), even in cases of low tumour content. Additionally, by sequencing introns, these assays can also identify *FGFR2* rearrangements with both known and novel partners, reducing the amount of tissue required for multiple analyses [17]. However, DNA-based assays do not provide information on the actual expression of the observed rearrangement at the mRNA level [18], and they can be challenging when dealing with large intronic regions [19]. Moreover, a genomic breakpoint occurring in an atypical intron, such as outside intron 17 for *FGFR2,* can result in the rearrangement being overlooked. For these reasons, it is highly recommended to complement a DNA-based analysis with an RNA-based analysis. This partner-agnostic approach provides direct information on the reading frame of the fusion, and the splicing out of introns in RNA overpass the issues with large introns observed in the DNA-based technique. However, with this approach, only SNVs expressed at the mRNA level can be identified, and sensitivity may be reduced in the case of low-quality RNA. Paraffin embedding, the most commonly used technique for tissue preservation, can contribute to RNA degradation [20]. Overall, these two tissue approaches are complementary and provide valuable insights into the characterization of BTC.

Even though tissue biopsy has long been considered the gold standard for the molecular analysis of BTC, it has a high risk of failure due to insufficient tumour content [8], which has sparked a growing interest in liquid biopsy for BTC [9]. Liquid biopsy, specifically the analysis of circulating DNA (ctDNA), shed by different lesions, provides an approximation of the spatial heterogeneity [21]. This means that ctDNA reflects not only the primary tumour and potential metastasis but also the intra-tumour heterogeneity. Additionally, due to its low invasiveness, serial ctDNA analysis can be performed. This enables the identification of resistance mechanisms [22] and provides valuable insights into the temporal heterogeneity of the disease [23].

If the diagnosis of CCA is suspected, the initial biopsy should include sufficient material for genomic analysis by NGS (10–15 slides are required) and ideally frozen material for *FGFR* fusion transcripts by RNA sequencing. It is also possible to perform a liquid biopsy, i.e., a blood sample to isolate circulating tumour DNA (ctDNA), which can also be helpful, even though this technique is less sensitive, especially in cases of low tumour burden. Molecular analysis on tumour tissue should not be replaced by analysis on ctDNA [7].

Interestingly, the mutational profile is different according to the primary tumour site. As already said, targetable mutations are more likely to be found in iCCA, *FGFR2* fusions and *IDH1* mutations in 15–20% of iCCA. In eCCA, the main alterations found, such as *KRAS* and *TP53,* are non-targetable mutations. Finally, *HER2* overexpression or mutation are overrepresented in gallbladder cancers [24]. Also, the molecular characteristics of BTC appear to vary according to the geographic and ethnic origin of patients. Patients of Asian origin have a higher rate of mutations in DNA repair genes and a higher tumour mutational burden, compared with Western, predominantly Caucasian, patients. Targetable molecular alterations, oncogenic drivers, are over-represented in Western patients. These differences could be due to different causes in Asia (liver fluke, hepatitis B virus) and in Western populations (metabolic syndrome) [25,26,27]. 

Molecular profiling-guided treatment of BTC has been shown to bring clear clinical benefits to patients [11,28]. In the French MOSCATO-01 trial, 43 patients with pre-treated advanced BTC were included, 23 of whom had a targetable molecular abnormality and were treated accordingly. The risk of death of the 23 patients with treatment matched to a molecular alteration was reduced by 70% compared with the 20 other patients [11].

### 1.1. FGFR Fusions 

Fibroblast growth factor receptors (FGFRs) are a family of four tyrosine kinase receptors (FGFR1 to 4), encoded by genes of the same name, involved in various cellular mechanisms via several signalling pathways, such as PI3K-AKT or RAS-MAPK pathways [29]. Oncogenic alterations of this family (amplifications, rearrangements, fusions or mutations) are identified in 7% of all solid tumours and associated with abnormally activated FGFR, resulting in cell proliferation, angiogenesis and immune escape [29,30]. The *FGFR* alterations identified in CCA are mainly represented by fusions involving *FGFR2*, associated with various partners, the most frequent being *BICC1* (nearly 30% of cases), found in 10–15% of iCCA [17,31,32,33,34]. iCCA with *FGFR2* fusion have been reported to be associated with female gender, younger age and the presence of concomitant *BAP1* mutation [17,35]. A favourable prognosis associated with *FGFR* alterations has been reported in the absence of specific treatment [36]. In a large retrospective cohort, the median overall survival (mOS) was 24.9 months vs. 14.8 months for iCCA patients with and without *FGFR2/3* alteration, respectively [37]. 

In routine care practice, the search for gene fusion is mainly carried out by NGS techniques, using targeted panels and RNA sequencing-based analysis. Immunohistochemistry (IHC) and fluorescence in situ hybridization (FISH) are no longer the methods used in first intention to identify gene fusion, because of a lack of sensitivity [2,38]. Detection of fusions including *FGFR2* from non-invasive blood liquid biopsies can be used for NGS (ctDNA), although the sensitivity remains lower compared to the methods using tissue samples [38].

Several specific competitive tyrosine kinase inhibitors (TKIs) of FGFR have been developed in recent years. An oral molecule targeting FGFR 1 to 3, pemigatinib, have recently received approval for use by the Food and Drug Administration (FDA) [39]. Pemigatinib monotherapy was evaluated in the multicentre phase II FIGHT-202 trial, demonstrating an objective response rate (ORR) of 37%, in a cohort of 147 patients with pre-treated CCA bearing *FGFR2* fusion or rearrangement (metastatic: 82%, third line or higher: 39%) [13]. The final results were reported at the ESMO World Congress on Gastrointestinal Cancer, with a median progression-free survival (mPFS) of 7 months in patients with *FGFR2* fusions/rearrangement and 2.1 months in patients with other *FGFR2* alterations. The mOS was 17.5 months, and reached 30.1 months in responder patients [13,40]. The safety profile was acceptable with the most common adverse event (AE) being hyperphosphatemia observed in 58% of patients, rarely symptomatic and controlled with dietary measures and phosphate binders. Alopecia and diarrhoea occurred in almost 50% of patients. Ocular toxicity, specific to FGFR inhibition, was observed in 28% of patients (including 1% with grade ≥ 3). Grade ≥ 3 AEs were observed in 69% of patients, mainly represented by hyperphosphatemia (14%), stomatitis, arthralgias, hyponatremia and fatigue. AEs leading to the discontinuation of treatment were reported in 10% of patients [13,40]. Pemigatinib is currently under evaluation for first-line treatment in the FIGHT-302 phase III trial in comparison with standard gemcitabine cisplatin (NCT036536). The development of another specific FGFR2 inhibitor, infigratinib, was finally halted. It had shown good results in the phase II trial, with a 23% response rate and a PFS of 7.3 months in 108 pre-treated FGFR2 fusion-positive BTC patients [41].

Despite these positive results, resistance to these treatments develops rapidly, due to secondary mutations in the kinase domain of FGFR2, which prevents the inhibitors from binding to this level [42,43]. 

Mutations in the kinase domain of FGFR are the most common mechanism of resistance to FGFR-TKI. Among others, gatekeeper mutations are mutations encoding amino acids located in the hinge region of the ATP-binding pocket of kinases. They play a role in controlling TKI access to the hydrophobic ATP-binding pocket and promote active kinase conformation by stabilizing the hydrophobic spine. The consequences of such mutations are the abolition of specific molecular interactions between receptor and inhibitor (by steric hindrance with substitution of one amino acid by another), or conformational changes in the receptor that would destabilize a strong inhibitor–receptor interaction or stabilize a weak inhibitor–receptor interaction. Some mutations involve an amino acid triad defining a molecular brake, in the hinge region of the kinase, which maintains the kinase in an inactive conformation. The mutation of these residues leads to uncontrolled activation of the kinase. Mutations may also involve alternative signalling pathways, such as the MAP-kinase pathway with RAS mutation, for example, bypassing FGFR inhibition [17,22,43,44,45,46,47,48,49] (Figure 1).

Thus, more powerful TKIs have been developed such as futibatinib (TAS-120), an irreversible tyrosine kinase inhibitor targeting FGFR1–4, approved by the FDA in September 2022 based on preliminary results of the FOENIX-CCA2 study, recently confirmed after an extended follow-up [39,50]. In this phase II trial involving 103 patients, futibatinib was associated with a 42% ORR and tumour control rate of 82.5%, an mPFS of 9 months and an mOS of 21.7 months. Hyperphosphatemia was the most frequent AE, and treatment was discontinued due to treatment-related AEs in only 4% of patients. Futibatinib has shown efficacy after progression with first-generation FGFR inhibitors as pemigatinib [50]. This molecule therefore represents a new therapeutic option in these patients and is currently being evaluated in first-line treatment in a phase III trial (FOENIX-CCA3 trial; NCT04093362). One of the current limitations of these ongoing phase III trials (FIGHT-302, FOENIX-CCA3) is the difficulty in recruiting patients. This can be explained by the low incidence of FGFR2-positive BTC. Moreover, the positioning of these trials is complex. Clinicians currently have a good first-line treatment for BTC, regardless of molecular profile, with the combination of gemcitabine cisplatin durvalumab. Usually, patients are referred to an expert centre after failure of the first-line treatment for molecular profiling and inclusion in a therapeutic trial if possible. For these first-line trials, it is now necessary to assess the molecular profile before any treatment, and to refer the patient to a centre where the trial is open, which in practice is not so simple. The health authorities are awaiting the results of phase III trials before granting definitive approval for these drugs.

New-generation inhibitors are in development, such as RLY-4008, a specific *FGFR2* inhibitor, retaining anti-tumour activity against the main resistance mutations. In the phase I/II ReFocus trial, in 116 patients (including 91 patients with CCA), anti-tumour activity was observed in advanced CCA and solid tumours with partial response in 74 patients (64%), and with a disease control rate (DCR) of 72%. In patients with CCA with *FGFR2* fusion/rearrangement naïve for FGFR inhibitors, the ORR was 52% (13/25), and in those treated at the recommended dose for the phase II study, the ORR was 100% (4/4). In patients with CCA with *FGFR2* fusion/rearrangement pre-treated with FGFR2 inhibitors, the ORR was 14% (7/50) and DCR was 80% (40/50). The safety profile was favourable with most frequent AEs represented by palmar-plantar erythrodysesthesia (57%) and stomatitis (56%), reflecting a more selective inhibition [51,52].

Interesting results were also reported with gunagratinib, an irreversible pan-FGFR inhibitor. In the phase II study, 17 included CCA with *FGFR2* fusion/rearrangement, and the ORR was 52.9% (9/17). The disease control rate (DCR) was 94.1% (16/17), and the median PFS was 6.93 months [53].

### 1.2. NTRK Fusions

The Tropomyosin Receptor Kinase (TRK) receptor is a transmembrane receptor with tyrosine kinase activity, present in adult neurons and playing a role in the physiology and regulation of the nervous system. The binding of a ligand from the neurotrophin family (nerve growth factor (NGF), brain-derived growth factor (BDGF) and neurotrophin 3 (NT-3)) to this receptor leads to its dimerization, followed by a phosphorylation cascade leading to activation of the MAP kinase, phosphoinositide phospholipase C or PI3K-AKT pathways, responsible for cell growth and differentiation, or inhibition of apoptosis. There are three classes of TRK receptors, TRKA, TRKB and TRKC, encoded by three genes, Neurotrophic Tropomyosin Receptor Kinase 1 to 3 (NTRK1, NTRK2 and NTRK3), respectively. For several years now, fusions between the 3′ part of the NTRK gene and the 5′ part of another gene have been described, resulting in the formation of a TRK fusion protein responsible for the constitutional activation of the receptor and pro-oncogenic phosphorylation cascades responsible for the development of numerous cancers [54].

In BTC, as in several other cancers, NTRK gene fusions have a low frequency, at less than 1% [55,56]. Two specific NTRK inhibitors, larotrectinib and entrectinib, have demonstrated efficacy in several phase I/II trials, with response rates ranging from 57 to 79%, including 7% to 16% complete responses in a wide range of NTRK-positive tumours [57,58]. In these trials, three patients with CCA were included, and a partial response was observed in two of them [57,58,59]. These two treatments have been approved by the FDA on the basis of an agnostic approach, for the treatment of all NTRK fusion-positive tumours [39].

Other fusion genes involving *ROS1* and *ALK* have also been described at low frequencies in CCA, and these patients are also likely to benefit from treatment with specific TKIs [34,35,60].

### 1.3. Isocitrate Dehydrogenase 1 (IDH1) Mutation

IDH1 and IDH2, encoded by eponym genes, are enzymes found in human cell cytoplasm and mitochondria, respectively. They catalyse the decarboxylation of isocitrate to alpha-ketoglutarate (α-KG). The mutant IDH1 and 2 proteins catalyse the conversion of α-KG to 2-hydroxyglutarate (2-HG). The oncometabolite 2-HG inhibits α-KG cofactor-dependent reactions. This leads to increased DNA methylation and impaired cell differentiation, promoting proliferation of liver progenitor cells with the transformation into cholangiocarcinoma cells [61].

Approximately 15% of iCCA carry an *IDH1* mutation. The IDH1-specific tyrosine kinase inhibitor ivosidenib showed a progression-free survival benefit against a placebo (mPFS = 2.7 months vs. 1.4 months, *p* < 0.0001), and a significant overall survival benefit was observed with an mOS of 10.3 months (95% CI, 7.8–12.4 months) with ivosidenib versus 7.5 months (95% CI, 4.8–11.1 months) with a placebo (HR, 0.79 (95% CI, 0.56–1.12); *p* = 0.09). When adjusted for crossover, the mOS with a placebo was 5.1 months (HR = 0.49 *p* < 0.001) [14,62]. Ivosidenib allows disease stabilization with an ORR of 2% and a DCR of 51%. This drug is well tolerated, with only 11% of side effects requiring dose reduction or discontinuation in the ClarIDHy trial. Based on these results, ivosidenib is currently recommended in *IDH1*-mutated pre-treated advanced iCCA (Table 1) [2].

### 1.4. HER2 Mutations and Amplifications

Human Epidermal Growth Factor Receptor 2 (HER2) is a protein of the tyrosine kinase family encoded by the *ErbB2* gene. Its heterodimerization with the HER3 receptor typosine kinase leads to the activation of several signalling cascades involved in oncogenesis, such as the MAP kinase pathway and PI3K-AKT. Overexpression or overactivation of the HER2 receptor mediated by mutation or amplification of the *erbb2* gene is therefore an oncogenic driver.

*ErbB2* mutations and amplifications are found in 10 to 15% of gallbladder cancers, in 5% of eCCA and less frequently in iCCA (1%) [8,24,69,70]. HER2 overexpression analysis is performed via immunohistochemistry. In the case of overexpression (score 2+ or 3+), fluorescent in situ hybridization (FISH) should be performed to confirm the result.

In the case of *ErbB2* mutation or amplification, patients with metastatic BTC could be treated with a HER2 inhibitor. The non-randomized phase II basket trial “My Pathway” evaluated the combination of trastuzumab and pertuzumab—two anti-HER2 monoclonal antibodies—with a response rate of 23% [65] (Table 1). Recent results from a phase II trial, presented at ASCO 2022, suggest that trastuzumab-deruxtecan—an anti-HER2 antibody conjugated with a topoisomerase I inhibitor—in unresectable HER2-overexpressing BTC is effective with a response rate of 36% and a disease control rate of more than 80% [66] (Table 1). Trastuzumab-deruxtecan has already been validated in the management of HER2-overexpressing breast cancer [71], and it is currently undergoing phase III evaluation in HER2-overexpressing gastric cancer with very encouraging results [72,73].

Another type of anti-HER2 drug has recently been developed: a bispecific antibody named zanidatamab. Zanidatamab is a humanized, bispecific monoclonal antibody directed against two non-overlapping domains of HER2. In the very recent phase IIb “HERIZON-BTC 01” study, zanidatamab demonstrated meaningful clinical benefit with a manageable safety profile in patients with treatment refractory, HER2-positive BTC. In patients with a HER2 score of 2+ and 3+, which represented 80 out of the 87 patients included, the ORR was 41.3% with a median response time of 12.9 months. No response was observed in the seven patients with a HER2 score of 0 or 1+. The mPFS in this study was 5.5 months (95% CI, 3.7–7.2), and the rate of OS at 9 months was 70% (95% CI, 57.8–79.1%) [67].

### 1.5. KRAS G12C Mutation

RAS proteins (including KRAS) are small GTPases. They are activated by membrane tyrosine kinase receptors for growth factors. RAS proteins are a family of proteins with a protooncogenic role through their involvement in numerous signalling pathways such as the MAP kinase and PI3K-AKT pathways.

*KRAS* mutations are common in digestive cancers (44%) and particularly in BTC [74]. In most cases, these mutations are undruggable. The *KRAS* G12C mutation is an exception, which can be targeted by a specific inhibitor, such as sotorasib or adagrasib. These two drugs are specific inhibitors of KRAS G12C, binding irreversibly by covalent bonding to the unique cysteine of KRAS G12C. The frequency of the *KRAS* G12C mutation is 4% in all digestive cancers and approximately 1% in BTC [74]. In the phase I CodeBreaK100 study, which included 129 patients with *KRAS*G12C-mutated tumours treated with sotorasib, 1 patient had biliary tract cancer for whom the best response was stable disease [75]. The efficacy of adagrasib has been demonstrated in metastatic non-small cell lung cancer [76] and metastatic colorectal cancer [77]. Recent results from the phase II Krystal-1 trial showed a 100% disease control rate and a 40% response rate in advanced non-colorectal *KRAS* G12C-mutated gastrointestinal cancers treated with adagrasib [68]. Of these, four out of eight patients (50%) with advanced BTC had a partial response (Table 1).

### 1.6. BRAF Mutation

The MAP kinase pathway (including the RAS-BRAF-MEK-ERK protein kinases) is involved in cell proliferation and survival. *BRAF* mutation, leading to the constitutive activation of this pathway, is notably involved in the process of oncogenesis of some melanoma, non-small cell lung cancer or colorectal cancer, for which BRAF inhibitors are validated in combination [78,79,80]. To date, more than 50 *BRAF* mutations have been described, mainly with the *BRAF* V600E mutation proving to be deleterious [81].

*BRAF* mutations are rare in BTC and are found almost exclusively in iCCA, with a prevalence of around 5%, including 1.5% *BRAF* V600E mutation [81].

The phase II “VE basket” trial evaluating the BRAF TKI vemurafenib in *BRAF*-mutated metastatic cancers showed mixed results with a response rate of 12% (one patient out of eight) in *BRAF*-mutated metastatic cholangiocarcinoma [82]. The dual blockade of BRAF and MEK with dabrafenib and trametinib, an effective strategy in *BRAF*-mutated metastatic melanoma, improved these results with a response rate of 47% (20 responses out of 43 patients) at the interim analysis of the phase II ROAR basket trial [63], leading to the FDA approval of dabrafenib-trametinib combination in *BRAF*V600E mutated advanced tumours, including BTC [39] (Table 1).

### 1.7. Mutation of Genes Involved in DNA Repair

#### 1.7.1. Mismatch Repair System (MMR)

The MMR system is a DNA repair system that is now well described, involving four proteins, MLH1, MSH2, MSH6 and PMS2, encoded by genes of the same name. It recognizes and repairs DNA mismatches that occur during replication. In the case of a deficiency in the MMR system (MMR-d), there is an accumulation of mutations responsible not only for oncogenesis, but also for the production of numerous MMR-d tumours, making them more sensitive to immunotherapy. Mutations in one of the proteins of the MMR system may be constitutional, forming part of a lynch syndrome with a predisposition to cancer of the colon, ovary and bile ducts, among others. A tumour can also be MMR-d due to a double somatic event or hypermethylation of the MLH1 promoter.

The prevalence of MMR-deficient status (MMR-d) or microsatellite instability (MSI) in BTC does not exceed 5% with a discrete overrepresentation in iCCA [83]. As in other tumour types, MMR-d status confers a particular sensitivity to immunotherapy [84]. In the multi-cohort phase II KEYNOTE-158 trial evaluating pembrolizumab in patients with metastatic MSI non-colorectal cancer, the response rate of the 22 patients included in the BTC cohort was 41% [64] (Table 1). Pembrolizumab is FDA-approved for MSI tumours [39].

The determination of MMR phenotype via the immunohistochemistry technique is recommended at the time of diagnosis of locally advanced or metastatic disease. In the case of MMR-deficient phenotypes, the MSI status should be confirmed via PCR analysis [3].

#### 1.7.2. BRCA1/2 Mutations

The prevalence of *BRCA* 1 and 2 mutations in BTC is between 3 and 5% [85,86]. *BRCA* 2 mutations are five times more frequent than *BRCA* 1 mutations. A *BRCA* 1/2 mutation could confer a sensitivity to PARP (poly ADP ribose polymerase) inhibitors, based on the principle of synthetic lethality, as already demonstrated in other tumour types. Indeed, PARP inhibitors are widely used in the management of ovarian [87,88] and breast [89,90] cancers with BRCA1/2 mutations, but also in metastatic pancreatic cancer with BRCA1/2 mutations [91]. There are reports of patients with *BRCA1/2*-mutated BTC treated with PARP inhibitors [92], and the use of olaparib in the management of advanced BTC with a DNA repair gene defect is being evaluated in a phase II trial (NCT 04042831) [93] (Table 1).

#### 1.7.3. Other Genes Involved in DNA Repair: ARID1A, BAP1, ATM and ATR

Other genes also involved in DNA repair systems are frequently mutated in biliary cancers. For example, mutations in *ARID1A* are found in 5–10% of biliary tract cancers, BAP1 in about 10% of CCAi and *ATM* and *ATRX* in 3–5% of CCAi. These molecular alterations could lead to increased sensitivity of the disease to PARP inhibitors, such as olaparib [86,93,94].

### 1.8. Molecular Profile: New Approaches to Guide the Treatment of BTC?

Since the publication of the results of the TOPAZ-01 trial evaluating the combination of durvalumab and gemcitabine-cisplatin, checkpoint inhibitors in combination with chemotherapy have been indicated as a first-line treatment for BTC with or without a targetable molecular alteration. The recent results of the phase III study MK3475–966 also validate the combination of pembrolizumab with cisplatin and gemcitabine in the same indication. However, the benefit of adding immunotherapy to chemotherapy is of the order of one to two months in terms of overall survival and remains modest. For this reason, it would be interesting to find factors that predict response. A recent publication suggests that the molecular profiling of BTC would enable tumours to be classified into four clusters. The authors explain that specific BTC clusters have distinct clinical and biological features, and such clusters may provide opportunities for therapeutic development. Immune-related biomarkers indicative of an inflamed tumour immune microenvironment were elevated in cluster 1, which was enriched for TP53, KRAS and ATM mutations. They identify relationships between individual driver genes and certain immune-related features, including enhanced M2 polarization of macrophages in IDH1-mutated BTC (cluster 3) and low immune infiltration in BTC with FGFR2 fusions or rearrangements (cluster 4), providing initial evidence for combining the targeted inhibition of specific drivers to reprogram the tumour immune microenvironment in combination with systemic immunotherapy [95].

## 2. Conclusions

Unresectable BTC remains a disease with poor prognosis. Molecular profiling should be performed at the time of diagnosis of advanced or metastatic BTC to drive therapeutic decisions from the second line of treatment, as this strategy has clearly been shown to bring clinical benefit to patients.

Some treatments, such as ivosidenib in IDH1-mutated iCCA, immunotherapy in MSI BTC, FGFR inhibitors in FGFR2-positive iCCA or entrectinib in NTRK-mutated BTC, are an alternative to second- or third-line chemotherapy. The development of new targeted therapies offers the prospect of improved management for an ever-increasing number of patients.

Finally, in the absence of validated treatments, the identification of a targetable molecular alteration may allow access to therapeutic trans-organ trials (anti-HER2, anti-BRAF, anti-KRASG12C, olaparib, other anti-FGFR2, etc.).

## Figures and Tables

**Figure 1 cancers-15-04446-f001:**
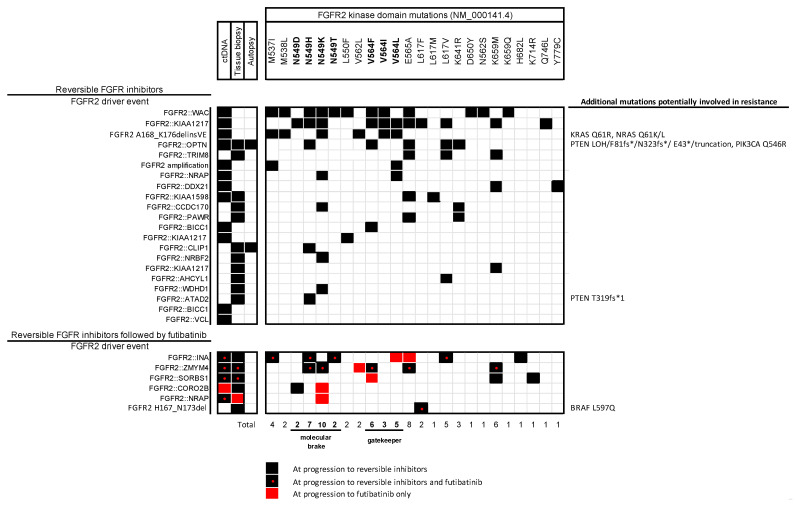
Illustration of the multiple resistance mutations to FGFR inhibitors. Patient-level data on molecular findings regarding resistance to selective FGFR inhibitors in *FGFR2*-driven cholangiocarcinoma, derived from the published study. The top panel includes patients progressing on reversible FGFR inhibitors (*n* = 21), while the bottom one includes patients receiving a sequence of reversible inhibitors followed by furtibatinib (*n* = 6). No published data regarding resistance in patients receiving upfront futibatinib are available thus far [17,22,43,44,45,46,47,48,49]. fs* = type of change is a frame shift.

**Table 1 cancers-15-04446-t001:** Summary of the main available treatments of biliary tract cancers with a targetable alteration.

Somatic Alteration	Primitive Tumour Location and Frequency	Study	Drug (s)	Results
*FGFR* fusion	15% of iCCA	FIGHT-202, [40] (phase II), ≥L2	Pemigatinib (FGFR 1–3 inhibitor)	ORR 37% DCR 82% mPFS 7 mo, mOS 17.5 mo
		FOENIX-CCA2, [50] (phase II), ≥L2	Futibatinib (Pan-FGFR inhibitor)	ORR 42%, DCR 83% mPFS 8.9 mo, mOS 20 mo
		ReFocus, [52] (phase I/II), ≥L2	RLY-4008 (Pan- FGFR inhibitor)	ORR 53% DCR 94% mPFS 6.9 mo
*IDH1* mutation	15% of iCCA	ClarIDHy, [14] (phase III) L2–L3	Ivosidenib	ORR 2%; DCR 51% mPFS 2.7 mo, mOS 10.8 mo
*BRAF* mutation	5% of iCCA	ROAR, [63] (phase II) ≥L2	Dabrafenib + Trametinib	ORR 42%
MSI	5% of BTC	Keynote-158, [64] (phase II multi cohorte) ≥L2	Pembrolizumab	ORR 40.9%
*BRCA1*/2 mutation	3–5% of BTC	NCT 04042831 (phase II, pending) ≥L2	Olaparib	Pending
*HER2* (mutation and amplification)	5% of eCCA 10–15% of GBC	My pathway, [65] (phase II basket) ≥L2	Trastuzumab + pertuzumab	11 pts with BTC, 8 amplifications, 3 mutations: ORR 3/8 and 1/3; mPFS 4.2 mo and 2.8 mo, respectively
		JMA-IIA00423, [66] (phase II) ≥L2	Trastuzumab-deruxtecan	ORR 36.4%; mPFS 5.1 mo, mOS 7.1 mo
		HERIZON-BTC 01, [67] (phase IIb)	Zanidatamab	ORR 41.3% mPFS 5.5 mo
*KRAS* G12C mutation	2% of *KRAS* mutations KRAS mutations: 20% of iCCA, 40–50% of eCCA	Krystal-I, [68] (phase II multi cohorte) ≥L2	Adagrasib	ORR 50%, mPFS 7.9 mo

BTC: biliary tract cancer; iCCA: intrahepatic cholangiocarcinoma; eCCA: extrahepatic cholangiocarcinoma; GBC: gallbladder cancer; L2–L3: second or third line of treatment; ≥L2: from second line of treatment; pts: patients; DCR: disease control rate; ORR: overall response rate; mPFS: median progression-free survival; mOS: median overall survival.

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
