# Peer review of "Targetable Molecular Alterations in the Treatment of Biliary Tract Cancers: An Overview of the Available Treatments"

_cancers, 2023, doi:10.3390/cancers15184446_

Round 1
Reviewer 1 Report
This is not a new topic as there have been some reviews on the progress of targeted therapies for BTC in the recent years (ex. PMID: 32171892, 35837710, 35839428, 3462656). There are also some other reviews covering of this topic in a broader theme (ex. PMID: 35032765, 36150578)
In this review, it is nice that the authors have listed a number of common gene alterations in BTC and summarized the results of clinical trials for drugs targeting these gene alterations. However, I hope the authors could make some more discussions over these trials. Like, what are the limitations of these trials? What can we learn from these trials? What are the future directions for the development of new targeted drugs or therapeutic strategies?
To make this review more informative, I would suggest the authors include a brief introduction for the molecular mechanisms (just like what’s written for IDH1/2) of each gene alteration in BTC. The working mechanism and of the targeted drugs for some of the gene alterations (like FGFR2 mutation) could also be explained in brief.
Some mistakes in the content:
1. Line 94-95: “The median progression free survival (mPFS) was 5.5 months in patients with FGFR2 fusions/rearrangement and 4.4 months in patients with other FGFR2 alterations.” According to reference 13, the mPFS was 6.9 months in patients with FGFR2 fusions/rearrangement and 2.1 months in patients with other FGFR2 alterations.
2. Line 108-109: PROOF trial for infigratinib has been terminated due to the discontinuation of infigratinib development in oncology within the sponsor's territory. Please update.
Line 121-122: I feel the sentence “Futibatinib has shown efficacy after progression with first-generation FGFR inhibitors as pemigatinib (35)” is better put before the sentence “This molecule therefore represents a new therapeutic option in these patients and is currently being evaluated in first-line treatment in a phase III trial (FOENIX-CCA3 trial; NCT04093362).”
Some minor mistakes in language:
Line 113-114: repeated “irreversible tyrosine kinase inhibitor targeting”.
Line 126: In sentence “…with partial responses in 74 (64%), …”, “patients” should be added before the comma.
Line 151: “founded” should be corrected as “found”, “cells” should be “cell”.
Line 160: “As well an overall survival significant benefit” should be corrected as “a significant overall survival benefit”.
Line 218: “exclusively iCCA” should be “exclusively in iCCA”.
Author Response
Dear Reviewer,
Thank you for your constructive comments and suggestions for improvement.
We have revised the manuscript to include analysis and discussion of ongoing therapeutic trials, particularly those involving FGFR inhibitors, which are highlighted in the new manuscript.
We add molecular mechanisms for different molecular alteration , and a paragraph on resistance mutations to FGFRi.
We have corrected the errors concerning the content and English language that you pointed out to us. Thank you very much for your help.
Thank you very much for the consiration given to our work.
Reviewer 2 Report
This study provides a comprehensive overview of the ongoing clinical approaches for the treatment of biliary tract cancer. The content is meticulously structured and enhanced by the inclusion of a table, facilitating a concise comprehension of treatment strategies.
Nevertheless, it is noteworthy that contemporary research is actively engaged in investigating the utilization of immunotherapy as a cornerstone for treating biliary tract cancer. Consequently, there is a requirement to systematically summarize this information and explore the research findings concerning the mechanistic operation of immunotherapy in the context of biliary tract cancer.
Author Response
Dear Reviewer ,
Thank you for your constructive comments and suggestions for improvement.
We've added a paragraph on what molecular profiling could contribute to immunotherapy treatment in biliary tract cancer (in particular, providing predictive factors for response). However, we have not elaborated on immunotherapy and its mechanism of action in all BTC, as our review focuses specifically on targetable molecular alterations in BTC.
Thank you very much for the consideration given to our work.
Reviewer 3 Report
This is a well-written review of genetic alterations and targeted drugs in biliary tract cancer. Please add the method for searching for genetic alterations in biliary tract cancer. Also indicate whether the genetic abnormality differs by race.
Author Response
Dear Reviewer,
Thank you for your constructive comments and suggestions for improvement.
We have added a paragraph on the method for searching for genetic alterations in biliary tract cancer, highlighted in the new manuscript. Also we have added information on distribution of molecular alterations according to patient ethnicity.
Thank you very much for the consideration given to our work.